# In-Situ Calibration of the Swarm-Echo Magnetometers

Robert M. Broadfoot[1], David M. Miles[1], Warren Holley[2], Andrew D. Howarth[2]

[1]Department of Physics and Astronomy, University of Iowa, Iowa City, 52242, United States
[2]Department of Physics and Astronomy, University of Calgary, Calgary Alberta, T2N 1N4, Canada

*Correspondence to*: Robert M. Broadfoot (robert-broadfoot@uiowa.edu)

**Abstract.**

CASSIOPE/e-POP, now known as Swarm-Echo, was launched in 2013 to study polar plasma outflow, neutral escape, and the effects of auroral currents on radio propagation in the ionosphere. The e-POP suite contains an array of eight instruments which include two fluxgate magnetometers on a shared boom. Until now, the two magnetometers relied on a set of prefight

calibrations which limited the accuracy of the magnetic field product and their utility for some applications. Here we present the results of an in-situ calibration performed on data from January 3, 2014, to January 30, 2021, and a case study showing the improvements the calibration has made to the data utility. Periodic vector-vector calibration using the Chaos magnetic field model results achieves an estimated RMS uncertainty of 9 nT during nominal operation. This data product is now openly available through the ESA Swarm repository.

## 1 Introduction

The CAScade Smallsat and Ionospheric Polar Explorer (CASSIOPE) containing the enhanced Polar Outflow Probe (e-POP) instrument suite (Yau and James, 2015) was launched in 2013 by the Canadian Space Agency in partnership with the University of Calgary, Communication Research Center in Ottawa, Magellan Aerospace, and MDA, the prime contractor for the mission. In 2018, the European Space Agency (ESA) adopted CASSIOPE into its Third Party Missions Programme and

inducted it into the Swarm constellation (Friis-Chistensen et al., 2008) as Swarm-Echo. The original Swarm spacecraft have roughly circular orbits at ~450 km for A and C and >500 km for B. e-POP was considered to be a desirable addition to the Swarm constellation because the orbit is highly elliptical with perigee at ~330 km and apogee at ~1500 km at launch. As such, it sweeps out a broader range of altitudes that Swarm A/B/C.

The scientific mission for the e-POP instrument suite is to study the Earth's ionosphere, thermosphere, and magnetosphere

while working to gain an understanding of plasma dynamics and their impact on radio propagation in the auroral ionosphere. The e-POP suite contains an array of eight instruments which includes two fluxgate magnetometers (MGF) (Wallis et al., 2015) separated by 32 cm center-to-center on a 92,9 cm carbon fiber boom (Figure 1). Until recently, MGF relied on a set of pre-flight calibrations discussed in Section 2 which limited the accuracy of the magnetic field product and their utility for some applications. Fluxgate magnetometers calibrations can evolve slowly over time, particularly due to baseline drift, and

the pre-flight calibrations cannot practically capture the stray fields from the spacecraft.

CASSIOPE is a three-axis stabilized spacecraft and uses reaction wheels to control the spacecraft attitude in a nominal +Z-to-nadir pointing mode(Figure 1) with magnetorquers used to periodically momentum dump the reaction wheels.. Originally, four reaction wheels were used to stabilize the spacecraft. However, in August 2016 one of the wheels failed and the remaining three wheels were slowed to compensate for this, and in February 2021 a second wheel failed which resulted in the remaining two being shut off while a solution to stabilize the spacecraft was investigated. Three months after re-acquiring a 3-axis stabilized attitude using two wheels, a third wheel failed in December 2021, forcing the spacecraft into a permanent spin-stabilized sun-pointing attitude. However, that time interval is beyond the scope of this manuscript.

Here we present the results of an in-situ vector calibration, performed to improve the accuracy of the MGF magnetometer data for the period comprising early mission through the failure of the second reaction wheel. We present the theory for vector magnetometer calibration, the limitations imposed by the existing pre-flight calibrations, the updates to the attitude determination software which were necessary for a successful calibration, the steps taken to select data for calibration, the results of the calibration over the entire mission length, a case study demonstrating the improved scientific utility of the MGF data, and future work planned to further improve the fidelity of the MGF data for the entire mission.

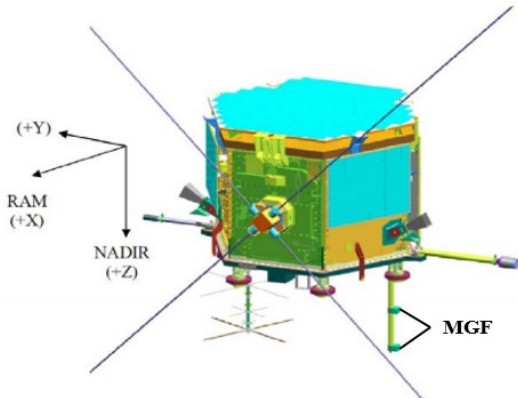

**Figure 1: The CASSIOPE spacecraft showing the two MGF magnetometers mounted at different distances from the spacecraft body on a common boom. The coordinate system shown represents the Common Reference Frame (CRF) with +Z pointing nadir, +X pointing ram, and +Y completing the right-handed coordinate system.**

## 2 Limitations of the pre-flight calibrations

Pre-flight calibration (Wallis, 2010) of the two MGF magnetometers was carried out from 2009-2010 at the Geomagnetic Laboratory of the Geological Survey of Canada at Anderson Road, Ottawa. An 8-foot Helmholtz Coil facility in Building 8 was used to cancel the main Earth field including local variations up to 4 Hz and apply various stimulus to characterize the sensitivity, orthogonality, instrumental zeros, and the rotation between the instruments These calibrations allow a reasonable reconstruction of the magnetic field vector (Wallis et al., 2015); however, they have several limitations. There was no

opportunity to attempt to estimate the stray magnetic field of the spacecraft prior to launch and, on orbit, zeros of the magnetometers are functionally the sum of the intrinsic instrumental zeros plus the static stray field of the spacecraft at the sensor position. The final deployment angle of the magnetometer boom could be estimated from a potentiometer in the joint but was insufficient to accurately rotate the measurements from the frame of each sensor to the Common Reference Frame (CRF) of the spacecraft and then into a geophysical frame. Additional tests were completed to estimate temperature dependence and the mutual interference of the two sensors. Mutual interference was found to exist between the two sensors due to the modest 32 cm of separation between them. On CASSIOPE, the instruments share a common clock which mitigates any interference from the respective drive signals making the interference primarily one of gain error from the stray fields generated by each sensor's magnetic feedback. The prefight interference was characterized by calibrating the sensors against a variety of fields while powering the other sensor on and off independently. Maximum interference was found to be in the +Z direction (Wallis, 2010) with the presence of a second sensor causing an error of approximately $-3.6 \times 10^{-4}$ nT/nT. On-orbit however, the sensors are never operated independently, so this effect is always present and is intrinsically captured by the instrument gains and then corrected by the application of the in-situ calibration. The presented in-situ vector-vector calibration was developed to resolve these issues and improve the absolute accuracy of the MGF data product.

## 3 Vector calibration of magnetometers

Here we describe the process we implemented to perform a full vector calibration of a three-axis magnetometer compared against a reference magnetic field. The vector calibration performed, and notation, is based on the method used by Olsen et al., (2003).

The presented vector calibration utilizes the full vector information by minimizing the vector residuals between the measured field and a model field. Specifically, we minimize $|\Delta B| = |B_{CRF} - B_{ref,CRF}|$ to obtain the calibration parameters. $B_{CRF}$ is the magnetic field vector in the CRF of the spacecraft and $B_{ref,CRF}$ is a reference field in the same frame. Before the vector residuals can be minimized, we must first characterize the relationship between the raw sensor data (in a slightly non-orthogonal reference frame) and the magnetic field vector. We assume that the raw sensor data have error in offset (**b**), sensitivity (**S**), orthogonality (**P**), and rotation ($R_A$).

Let **E** be the raw sensor data (in engineering units, eu) that is related to the magnetic field vector $B_{CRF}$ in the common reference frame by

$$E = SPR_A B_{CRF} + b \tag{1}$$

where,

$$b = \begin{pmatrix} b_1 \\ b_2 \\ b_3 \end{pmatrix} \tag{2}$$

is a vector of offsets (given in units of eu) comprising the superposition of the instrumental zeros and the static stray field of the spacecraft platform,

$$\mathbf{S} = \begin{vmatrix} S_1 & 0 & 0 \\ 0 & S_2 & 0 \\ 0 & 0 & S_3 \end{vmatrix} \tag{3}$$

is a diagonal matrix with each element representing a scale value often called sensitivity (given in units of eu/nT) for each magnetometer axis. Based on the results of the pre-flight calibrations (Wallis, 2010) we assume that electronic cross-talk between channels is negligible and hence the non-diagonal terms can be assumed to be zero.

$$\mathbf{P} = \begin{vmatrix} 1 & 0 & 0 \\ -\sin(u_1) & \cos(u_1) & 0 \\ \sin(u_2) & \sin(u_3) & \sqrt{1 - \sin^2(u_2) - \sin^2(u_3)} \end{vmatrix} \tag{4}$$

is a matrix that describes the projection of the magnetometer by angles $u_1, u_2, u_3$ (one for each axis pair) from a non-orthogonal frame into an orthogonal one. $\mathbf{R}_A$ is a rotation matrix (order '1-2-3' in the case of MGF) consisting of three separate Euler angles $e_1, e_2, e_3$ which describes the rotation between the magnetometer reference frame and the common reference frame. It should be noted that the choice of rotation order is largely arbitrary as long as it is able to fully describe the rotation from the instrument reference frame into the CRF. The rotation parameters do not have any effect on the magnitude of the calibrated field, however, but are necessary for the vector calibration to ensure the alignment of the frame of the sensor data and the reference field.

These 12 basic calibration parameters (3 offsets, 3 sensitivities, 3 orthogonalities, and 3 Euler angles allow us to find the magnetic field vector in the common reference frame from the sensor data using

$$\mathbf{B}_{CRF} = \mathbf{R}_A^{-1} \mathbf{P}^{-1} \mathbf{S}^{-1} (\mathbf{E} - \mathbf{b}) \tag{5}$$

with,

$$\mathbf{S}^{-1} = \begin{vmatrix} \dfrac{1}{S_1} & 0 & 0 \\ 0 & \dfrac{1}{S_2} & 0 \\ 0 & 0 & \dfrac{1}{S_3} \end{vmatrix} \tag{6}$$

and,

$$\mathbf{P}^{-1} = \begin{vmatrix} 1 & 0 & 0 \\ \dfrac{\sin(u_1)}{\cos(u_1)} & \dfrac{1}{\cos(u_1)} & 0 \\ -\dfrac{\sin(u_1)\sin(u_3) + \cos(u_1)\sin(u_2)}{w\cos(u_1)} & -\dfrac{\sin(u_3)}{w\cos(u_1)} & \dfrac{1}{w} \end{vmatrix} \tag{7}$$

Where $w = \sqrt{1 - \sin^2(u_2) - \sin^2(u_3)}$ and $\mathbf{R}_A^{-1} = \mathbf{R}_A^{T}$ from the properties of rotation matrices.

The twelve calibration parameters can now be obtained by minimizing the difference of the squared residuals

$$\left\|B_{CRF} - B_{REF,CRF}\right\|^2 \tag{8}$$

in a least squares sense.

Obtaining the parameters this way will involve solving a set of non-linear equations which will be dependent on initial guess parameters. However, following the procedure outlined in Olsen et al., (2020), equation (5) can be rewritten as

$$R_A^{-1}P^{-1}S^{-1}(E - b) = AE + \widetilde{b} \tag{9}$$

where $A = R_A^{-1}P^{-1}S^{-1}$ is a 3x3 matrix and $\widetilde{b} = -Ab$. This now allows the equation to be solved as a linear inverse problem which is no longer dependent on initial guess parameters.

The calibration parameters can then be determined by reforming the linearized results of $A$ into matrix form and decomposing using QL decomposition, which decomposes $A$ into two matrices: $Q$ and $L$. Here $Q$ is an orthogonal matrix and $L$ is a lower triangular matrix. There are different algorithms to perform this decomposition, here we use a Matlab function (Houtzager, 2022) that performs this task. This algorithm treats Q as a product of a series of elementary reflectors and uses these to reduce the matrix L to lower triangular form column by column (Parlett, 1998). Other methods of matrix decomposition yield different matrix forms (QR, LQ, LU, etc). However, QL decomposition matches the form of the matrices used to originally create $A$. As such, we can set $Q = R_A^{-1}$ and $L = P^{-1}S^{-1}$. From there, the three Euler angles can be obtained from the elements of $Q$. Since $L$ is a lower triangular matrix, which combines two separate matrices, we must use the knowledge that the three sensitivities must be positive, then the orthogonalities and sensitivities can be solved for using algebra. Lastly, the offsets can be obtained from $b = -A^{-1}\widetilde{b}$.

In addition to the twelve basic calibration parameters, other missions such as Cryosat-2 (Olsen et. al, 2020) and Ørsted (Olsen et. al, 2003) have had success expanding equation (9) to consider additional effects due to non-linearity and cross-talk and expanding individual terms to take temporal variations as well as effects from external sources such as temperature and stray current from the solar panels and batteries into consideration. These additional parameters may reduce outliers in the data and improve the overall fit with the reference field as well as reduce the variability of the individual calibration parameters. For this paper, however, we will focus on the improvements in the data fidelity from the 12 basic parameters only, with the inclusion of the additional terms and regularization being considered for future work.

## 4 Required updates to CASSIOPE attitude solution

An accurate attitude solution is critical to a successful vector calibration. It is used to rotate the reference magnetic field from a geophysical coordinate system into the coordinate system of the magnetometers. Though CASSIOPE has several methods to determine spacecraft attitude, the difference between the quality of attitude sources is large. Deficiencies were found in the original attitude determination software that impacted the overall quality of the attitude solution. This section will focus on the updates that were made to the attitude determination software with Section 5 detailing the method used to select quality attitude data and rotate the reference magnetic field into the magnetometer's coordinate system. The previous attitude

solution was used for versions 1.x.x of the data product whereas the improved solutions were used for 2.x.x. The current attitude file version of 1.3 should not be confused with the data product version. The Swarm-Echo attitude determination system is composed of a micro-Advanced Stellar Compass (µASC) with two camera heads provided by the Technical University of Denmark, six Adcole 46300 coarse sun sensors (CSS), and two Billingsley TFM-100S magnetometers. Fine attitude knowledge is achieved via the µASC, with the coarse sun sensors and magnetometers providing coarse solutions when a µASC solution is not available. The µASC is specified by the manufacturer to provide solutions with errors less than 2 arcseconds (3σ) until end of life (Jørgensen et al., 2003), though there is a known mounting angular offset of ~0.447 degrees to bring the two cameras into agreement, which is averaged across both cameras.. The attitude solutions provided by the Attitude Determination and Control System (ADCS) derived from CSS have a known error of up to 30 degrees (2σ) which makes them unsuited for calibrating or transforming the MGF instruments and these intervals are rejected during data processing. Early mission attitude data were generated only as Yaw-Pitch-Roll (YPR) values using the AGI Systems Toolkit (STK) from onboard pre-processed attitude telemetry from Swarm-Echo's ADCS. The original system merged the higher-accuracy solutions generated from the star-trackers with the low-accuracy solutions from the CSS and treated it as a continuous dataset. This caused STK to reject or discard large sections of the attitude solution or provide solutions with visible steps in the data when the system would transition between attitude determination sources. Reproducibility of these events was a challenge as there is limited visibility into the STK software. Early on, this was less of an issue as the cameras were still new. However, as the cameras aged and were exposed to radiation on-orbit the individual star tracker solutions began to diverge from each other, and increasingly more data began to be rejected. The degrading quality of the data necessitated a change to the attitude determination algorithms. The revised attitude solution included improved alignment between different star camera modes, corrections for orbital and annual aberration which are caused by the relative velocity of the stars within the camera heads and thermal effects in the star cameras. These corrections closely follow the procedures outlined in the Swarm Level 1b Processor Algorithms (SW-RS-DSC-SY-0002). Additionally, corrections were made to frame, location, and epoch transforms. The details of these corrections will be covered in a forthcoming publication by the CASSIOPE science operations team. Switching to a Spherical linear interpolation (Slerp) or a Spherical QUADrangle (SQUAD) (Shoemake, 1987) interpolation rather than per-element interpolation further improved the robustness of the attitude transform by enforcing continuity and smoothness of the attitude solution over multiple measurement points which is appropriate for a physical spacecraft moving in physical space.

## 5 Attitude selection and reference field rotation

In-situ calibration of the MGF instruments requires rotating the reference magnetic field from its native North, East, Center (NEC) frame into the local CRF frame of the spacecraft by convolution against the spacecraft attitude solution. As noted above, the spacecraft attitude solution is obtained from multiple sensors that must first be rotated from their native reference frames and merged into the CRF. Each primary attitude datum is provided by an individual µASC solution. The position of

the camera heads is on the Y-axis of the spacecraft (Figure 2), separated by 130° in the Y-Z plane, with the optical axis 25° from the X-Y plane (Figure 2).

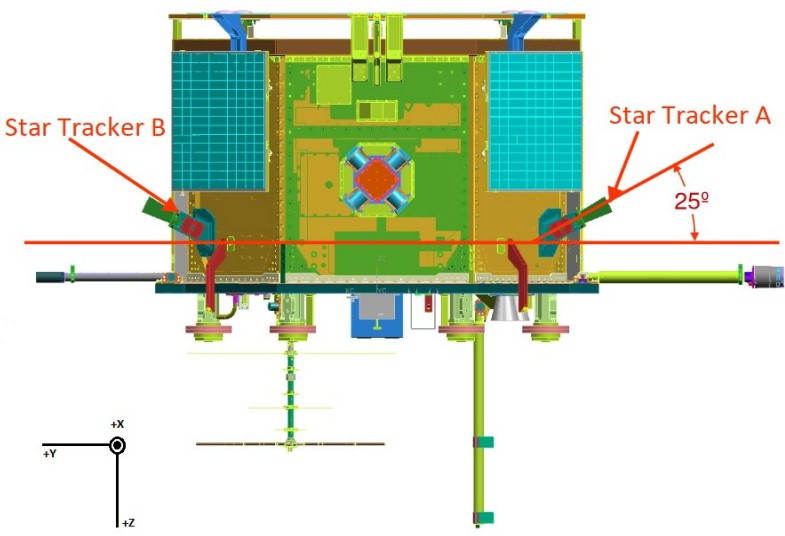

**Figure 2: Location and orientation of the two star-tracker camera heads that provide the e-POP attitude solution.**

The highest quality secondary-source solution is the onboard-merged 'dual' µASC solution, which is considered of lower quality to ground-processed camera solutions as the algorithm used to merge them onboard cannot be separated into the separate solutions for thermal correction. Following this, attitude solutions derived from the coarse sun sensors and bus magnetometer are considered. Solutions that use calibrated solar-vector and magnetic-field steering are considered first, followed by solutions using uncalibrated vector and field estimates. These solutions are used in fine-steering dropouts of

greater than 5 seconds, and 2 minutes respectively. Periods where there are no attitude sources available for extended periods of time (10 minutes) have zero-quaternion sentinel solutions placed surrounding the period. This is both to segment interpolation, and to limit the availability of incorrect solutions. The attitude solutions are then rotated from the coordinate frame of the source into CRF to bias interpolation towards the nominal operational attitude on CASSIOPE. These off-cadence 'definitive' solutions are then interpolated to 1Hz using SQUAD (Shoemake, 1987) quaternion interpolation. These

interpolated quaternions are then rotated into the International Terrestrial Reference Frame (ITRF) with a CRF-to-ITRF transform for publication in the Swarm-Echo attitude CDF product. The mathematics for the CRF-to-ITRF transformation will be detailed in the forthcoming publication mentioned in Section 4. Supplementary metadata are also derived from these definitive solutions, such as Time-To-Solution & Data-Source, to allow for the accuracy of both the interpolation and the raw data source to be measured and are included in the MGF data product. One additional rotation is performed for the final

MGF data product CDF which rotates the quaternions from ITRF to the North East Center (NEC) frame.

The process to perform this rotation involves generating the quaternions from the rotation matrix that describes the rotation from the ITRF position data into NEC and each matrix row is generated in reverse order (i.e. Center, East, North) and can be obtained by the following equations

$$Center \ = \ -\frac{r}{|r|} \tag{10a}$$

where $r$ is the ITRF position vector, then,

$$East = Center \ x \ (0,0,1) \tag{10b}$$

and finally,

$$North = East \ x \ Center \tag{10c}$$

which completes the right-handed coordinate system.

The matrix terms are then converted into quaternions $q_{NEC \rightarrow ITRF}$ and multiplied with the $q_{ITRF \rightarrow CRF}$ to make $q_{NEC \rightarrow CRF}$ quaternions which are included in the MGF data product and allow us to transform our reference magnetic field model, whose native coordinates are NEC, into the local CRF of the measured magnetic field data to enable vector-vector calibration.

Preliminary analysis suggests that the accuracy of the ephemeris is comparable to the values determined by Montenbruck et

al., 2019. The daily attitude files are in CDF format and contain the $q_{ITRF \rightarrow CRF}$ quaternions and the daily ITRF position files are in SP3 format which is a precise ephemeris format developed by the National Geodetic Survey to store orbit information. Both are publicly available at https://epop-data.phys.ucalgary.ca/ .

**6 Data selection and calibration**

As e-POP lacks an absolute scalar magnetometer, our in-situ calibration method requires that the data for calibration be compared against a reference magnetic field. We use the Chaos-7.7 field model (Finlay et al., 2020) which includes contributions from the core, lithospheric, and external (such as large-scale magnetosphere). We select data that falls within $\pm 55°$ geographic latitude, since the Chaos model does not contain terms to account for disturbances in the polar regions. However, before we perform the calibration, additional data selection and processing is performed to limit outliers in the

calibration.

Prior to in-situ calibration, we reduce the data from its native 160 sps to 1 sps to reduce the computational burden. We bin the data into seven-day intervals, which is an experimentally determined balance of sufficient data for a robust calibration while capturing time-varying effects. There are exceptions to this rule, especially prior to the first wheel failure when data coverage in non-polar latitudes was extremely sparse.

We have found that the quality of data selected to derive the in-situ calibrations is generally more important that the quantity of data. Consequently, we cull the data used for the calibration using information from the attitude, bus telemetry, and

location files, as well as conditions given by the Kp and Disturbance storm time (Dst) indices to identify intervals where the spacecraft signal is low, the geomagnetic field is undisturbed and hence well represented by the Chaos model, and we are far from likely auroral disturbances.

For calibration, we select data that fall within $\pm 55°$ latitude during geomagnetically quiet times. We consider geomagnetically quiet to be when the Kp index does not exceed 3 or the Dst index does not exceed a change of 3 nT per hour when the data were taken. From the attitude files we flag any data where the attitude solution was not generated by at least one of the star tracker cameras, due to the large (up to 30°) error in solutions generated from the CSS as mentioned in section 4. We also flag any data where the signal has dropped out for greater than ten seconds or there is greater than ten

seconds until the next signal is obtained due to potentially large errors when interpolating the attitude solution. Lastly, we flag any data where the rotation rate of the spacecraft exceeds 0.03 degrees/sec.

Figure 3 shows these flags as well as the source of the provided attitude solution and the effect they have on the data. The discrepancy of approximately -800 nT between solutions derived from a single star tracker camera (ST-A) and those derived from the CSS justify the exclusion of attitude solutions not derived from at least one star-tracker for the data used in

calibration. From the Bus telemetry files we flag any data where the magnetotorquers were engaged as they suddenly produce a 6000 nT step of stray field on axis at 1 m (Wallis, 2010) which exceeds the bandwidth of MGF, rendering the data clipped and unusable. It should be noted that even though the flagged data are excluded from the calibration process, all data and attitude flags are included in the daily MGF data product so the user can decide if the flagged data are useable.

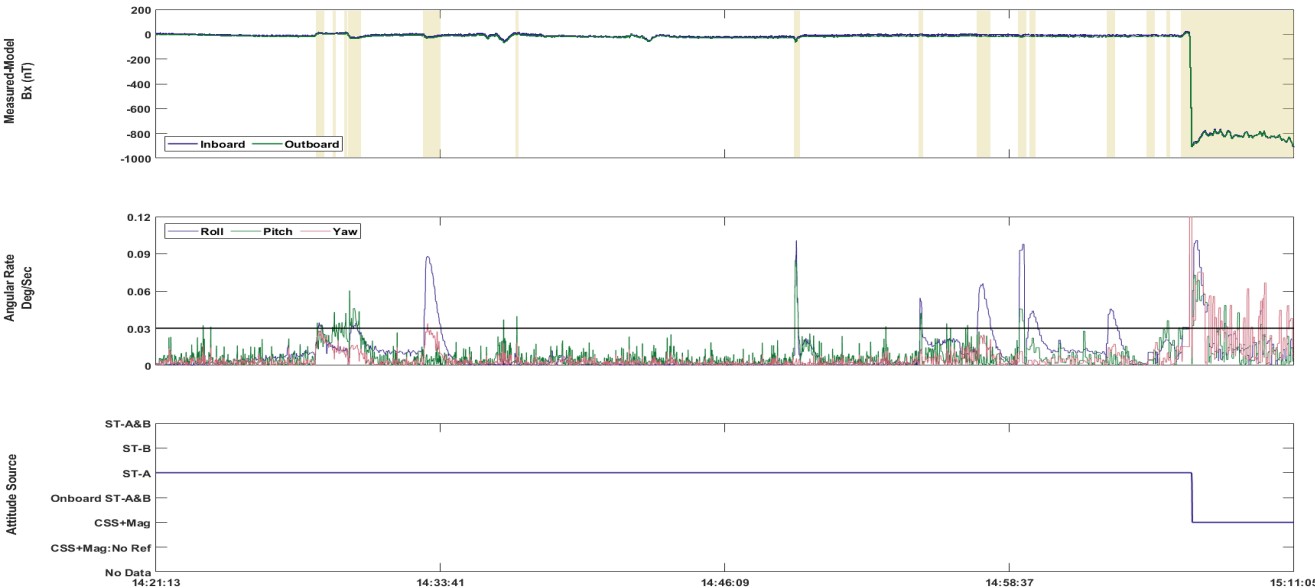

**Figure 3: (Top)Residuals for the $B_x$ component for inboard (blue) and outboard (green). Areas marked in yellow represent any point where the rotation rate exceeded 0.03 deg/sec or the attitude solution was not generated by at least one star-tracker camera. (Middle) The angular rotation rate corresponding to the data in the top plot. (Bottom) The seven potential attitude sources for the data which have significantly different accuracies of solution. The potential errors in the attitude solution derived from CSS versus one of the star-tracker cameras is clearly shown by the transition from ST-A (star tracker A only) to CSS (coarse sun sensor).**

After the extensive filtering described above, we use iteratively re-weighted least squares to minimize the difference in the vector residuals between the sensor data and the Chaos model for each seven-day interval.

$$d^T W d \tag{11}$$

Where **d** is the residual vector $B_{CRF} - B_{Chaos}$, containing all the selected data for the seven-day interval and **W** is a weight matrix. We use Huber weights (Huber, 1981) where the elements of the weight matrix are determined by the following criteria

$$W = \begin{cases} |r|^{-1} & r > 1 \\ 1 & r \leq 1 \end{cases} \tag{12a}$$

where **r** is determined from a combination of the residuals (**d**), the leverage (**h**), the median absolute deviation of the residuals (**s**), and a tuning constant (c) and is given by

$$r = d\left(cs\sqrt{1-h}\right)^{-1} \tag{12b}$$

with $c = 1.345$ (Holland and Welsch 1977) as it is less sensitive to outliers in non-polar latitudes.

## 7 Results/Discussion

Calibrations were performed for the period of January 3, 2014 to February 4, 2021, for all data up to the second wheel failure. This resulted in a total of 323 7-day calibrations after accounting for periods with sparse coverage or an insufficient amount of quality data for the calibration to converge. A mission-averaged static calibration was calculated from the 323 results and compared with the sensitivities and orthogonalities from the preflight calibrations (Table 1). No pre-flight estimates are shown for the rotations due to the potential variability of the boom deployment. The calculated pre-flight zeros (Wallis, 2010) are included for completeness. However, it should be noted that these zeros are not the same as the offsets calculated as part of the in-situ calibration as they only represent the intrinsic zero-offsets of the sensor and electronics electronics and do not include effects from stray field from the as it was not possible to quantify those prior to integration. The large offset in $b_1$ for the outboard sensor is likely due to a magnetized object on the boom near the sensor – the survival heater added to the boom during final integration is a potential candidate. Additionally, a sign error was discovered in the equation used to calculate the pre-flight orthogonalities, which accounts for the flipped sign of the values.

After accounting for the two discrepancies mentioned in the previous paragraph, the similarities between the two give us confidence in our mission averaged result. Any 7-day interval that lacked sufficient data for calibration after culling based on the criteria from section 6 will instead rely on the mission-averaged value to provide potential scientific utility to the usable data in those sets. Additionally, the calibration values that converged were compared against the mission average (Figure 4) results for orthogonality. We determined that jumps in orthogonality larger and +/- 0.1° would be non-physical as it would cause a catastrophic failure in the physical integrity of the sensor itself. This resulted in 22 (6.8 %) of the calibrations to be replaced with the mission-averaged values.

|  | In-situ Calibration | | Preflight Calibration | |
|---|---|---|---|---|
|  | Inboard | Outboard | Inboard | Outboard |
| $S_1$ [eu/nT] | 1.0044 | 1.0024 | 1.0044 | 1.0025 |
| $S_2$ [eu/nT] | 0.9979 | 1.0020 | 0.9984 | 1.0029 |
| $S_3$ [eu/nT] | 1.0503 | 1.0534 | 1.0473 | 1.0519 |
| $u_1$ [°] | 89.87 | 90.11 | 90.12 | 89.93 |
| $u_2$ [°] | 89.71 | 89.88 | 90.10 | 90.02 |
| $u_3$ [°] | 90.01 | 90.04 | 89.81 | 89.93 |
| $e_1$ [°] | 2.73 | 2.68 |  |  |
| $e_2$ [°] | -0.09 | 0.21 |  |  |
| $e_3$ [°] | 2.23 | 1.96 |  |  |
| $b_1$ [eu] | 1.47 | -199.41 | 1.6* | 2.5* |
| $b_2$ [eu] | 2.10 | 1.20 | 2.9* | 6.6* |
| $b_3$ [eu] | 8.33 | 24.22 | 3.6* | 3.5* |

**Table 1: Mission averaged values for the inboard and outboard sensors compared to the values obtained from the preflight calibrations. A sign error was discovered in the original matrix used to calculate the preflight orthogonalities which is why the discrepancy between them is so large. Orthogonalities are given in degrees and displayed as 90+Orthogonality. The large offset for Outboard $b_1$ is likely due to a magnetized object on the boom near the sensor. The pre-flight values marked with '*' represent the instrument zeros, or the offsets that would be measured by the magnetometer in zero magnetic field and are not the same as the in-situ offsets which include effects from stray-field from the spacecraft.**

Every rejected calibration set occurred prior to the first wheel failure in August 2016. While the sparsity of the data likely plays a significant role in the variability, the values show a noticeable decrease in variability in the calibration set immediately following the loss of the first wheel as seen in Figure 4. The larger variability in the outboard sensor is likely due to the large offset seen in the $b_1$, and to a lesser degree $b_3$ components. As the numbers are much larger in comparison to the other results, the variation in those values is also proportionally larger. It should also be noted that maximum data coverage did not occur until the later years in the mission. This implies that the reaction wheel tone has a significant impact on the calibration results and that steps will need to be taken to mitigate the wheel tone in future data releases and doing so should have a significant impact on the calibration results for the early mission.

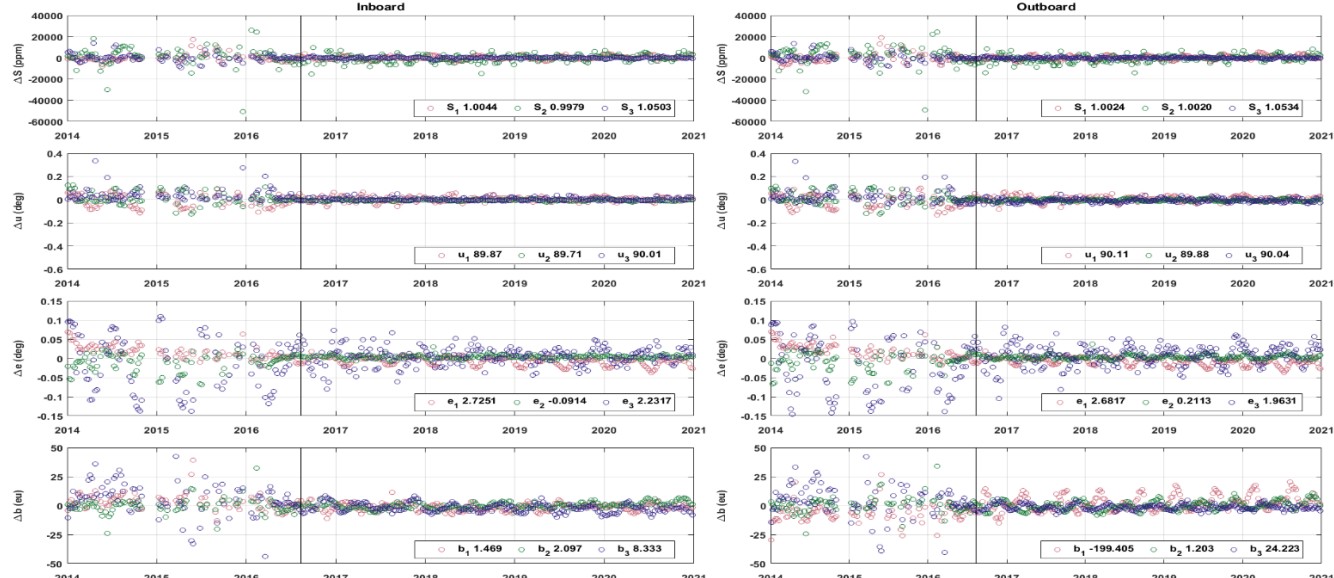

**Figure 4: Deviation of individual calibration results from the mission average. Mission-averaged values shown in legends. The greatest amount of variability in the results occurs prior to the first reaction wheel failure in August 2016 which is denoted by the black vertical line in each plot. Periodic behavior seen in the results is planned to be mitigated in future data releases using regularization.**

To further validate the calibration results we calculated the residuals for all 1-Hz data compared to the Chaos model for the calibration set until the start of 2021 (Figure 5). The results show, as expected, that the residuals in the non-polar latitudes (+/- 55°) are small and increase in the auroral latitudes. The darker area in the residuals represents the rms error for the residuals at that degree of latitude.

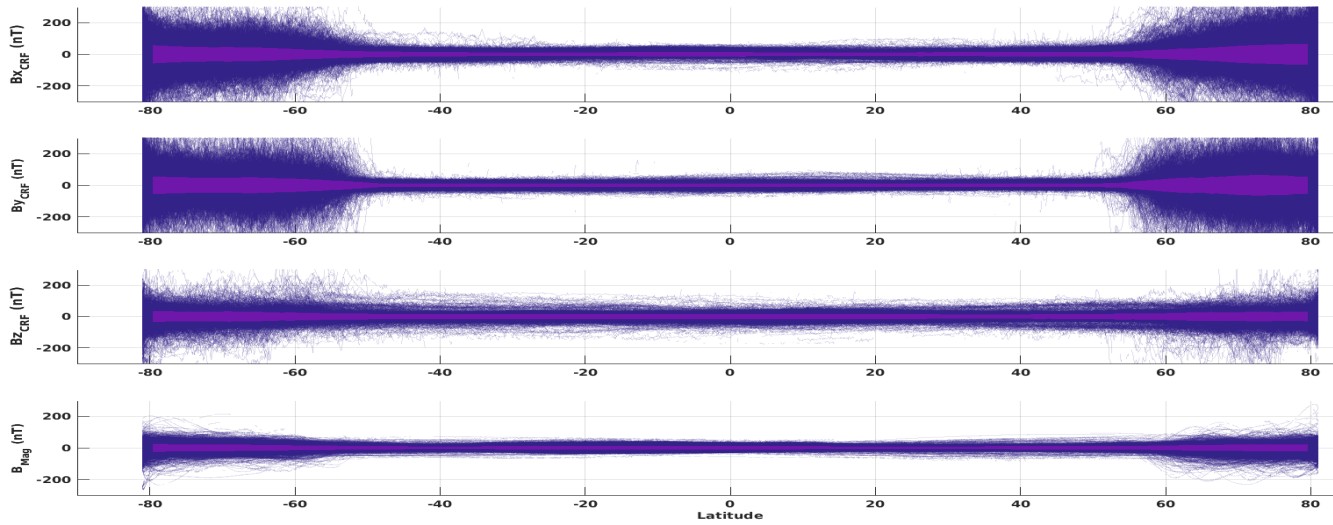

**Figure 5: All calibrated 1-Hz residuals for the outboard sensor with the Chaos model versus latitude. The same data culling as in section 6 was used for this plot and any data that did not meet that criteria was not included. The darker section in the data represents the RMS error for the data binned by each degree of latitude.**

We then calculated the average residuals and average rms error for non-polar latitudes by year (Table 2) to see to what degree the increased data coverage and loss of the first reaction wheel affected the results. As expected, the largest improvements in the averaged residuals occurred between 2015 and 2017 with the average residuals and rms error steadily improving each year. We expect to improve on these initial numbers in future data releases by adding additional terms to the calibration such as temperature and major spacecraft bus currents.

| Inboard | | | | | | | | | | | | | |
|---|---|---|---|---|---|---|---|---|---|---|---|---|---|
| 2014 | | 2015 | | 2016 | | 2017 | | 2018 | | 2019 | | 2020 | |
| avg | rms | avg | rms | avg | rms | avg | rms | avg | rms | avg | rms | avg | rms |
| $B_x$ -2.80 | 13.97 | -2.91 | 14.61 | -0.21 | 13.73 | 0.61 | 13.23 | 0.40 | 12.79 | 0.33 | 11.65 | 0.26 | 11.58 |
| $B_y$ 0.86 | 20.34 | -0.81 | 22.13 | 0.15 | 11.19 | -0.04 | 10.64 | 0.16 | 10.30 | 0.27 | 9.91 | -0.14 | 9.32 |
| $B_z$ -1.53 | 10.71 | 0.75 | 12.12 | -0.18 | 11.26 | 0.09 | 10.77 | -0.17 | 10.84 | 0.03 | 11.11 | 0.00 | 10.67 |
| \|**B**\| -2.80 | 16.21 | -2.91 | 19.23 | -0.21 | 9.23 | 0.61 | 9.76 | 0.40 | 9.05 | 0.33 | 8.81 | 0.26 | 8.85 |
| Outboard | | | | | | | | | | | | | |
| 2014 | | 2015 | | 2016 | | 2017 | | 2018 | | 2019 | | 2020 | |
| avg | rms | avg | rms | avg | rms | avg | rms | avg | rms | avg | rms | avg | rms |
| $B_x$ 2.87 | 17.12 | 3.63 | 18.73 | 0.46 | 13.58 | 0.53 | 12.99 | 0.42 | 12.67 | 0.27 | 11.71 | 0.07 | 11.61 |
| $B_y$ 1.27 | 14.70 | -1.29 | 14.92 | 0.17 | 11.22 | -0.10 | 10.48 | 0.28 | 10.14 | 0.25 | 9.80 | -0.10 | 9.17 |
| $B_z$ -0.52 | 19.91 | 1.59 | 21.99 | -0.39 | 11.00 | 0.13 | 10.40 | -0.23 | 10.41 | 0.02 | 10.64 | -0.01 | 10.15 |
| \|**B**\| 2.87 | 11.35 | 3.63 | 13.28 | 0.46 | 8.81 | 0.58 | 9.34 | 0.42 | 8.57 | 0.27 | 8.45 | 0.07 | 8.33 |

**Table 2: Average residuals and average RMS error for non-polar latitudes by year. The largest change in average residuals occurred between 2015 and 2017, coinciding with the loss of the first reaction wheel and subsequent slowing of the remaining wheel rates. The steady improvement in the results every year after can be attributed to the increased data coverage.**

To show improvements made to the data utility, we present a case study of a recreation of Figure 1 from Miles et al. 2018 (Figure 6) using the newly calibrated data and improved attitude/location data. The original publication used the spacecraft cross-track $B_y$ component as the calibration and attitude solution were not sufficient to use the geophysical data. It can be immediately seen that the presumed non-physical features between 62° and 64° geographic latitude in the original figure were removed while features between 64° and 66° geographic latitude that correspond with aurora visible in the superimposed image were retained. The improved attitude solution and calibration successfully mitigate non-physical magnetic variation due to platform motion while preserving legitimate changes in in-situ geophysical field. This plot combines the calibrated MGF data for the cross-track magnetic field along the path of the spacecraft from the outboard sensor (to reduce the effects of stray field from the spacecraft) with the Fast Auroral Imager (FAI). The black dot at 06:49:36 represents the position of the spacecraft when the overlayed auroral image was taken, and all other magnetic field readings are not associated with that image. While the purpose of this case study is to show the initial improvements made to the MGF data, it also shows one of many possible ways that scientific data from the instruments aboard Swarm-Echo can be combined to explore different events. In addition to the MGF data product, data for the FAI and other instruments are also publicly available at https://epop-data.phys.ucalgary.ca/

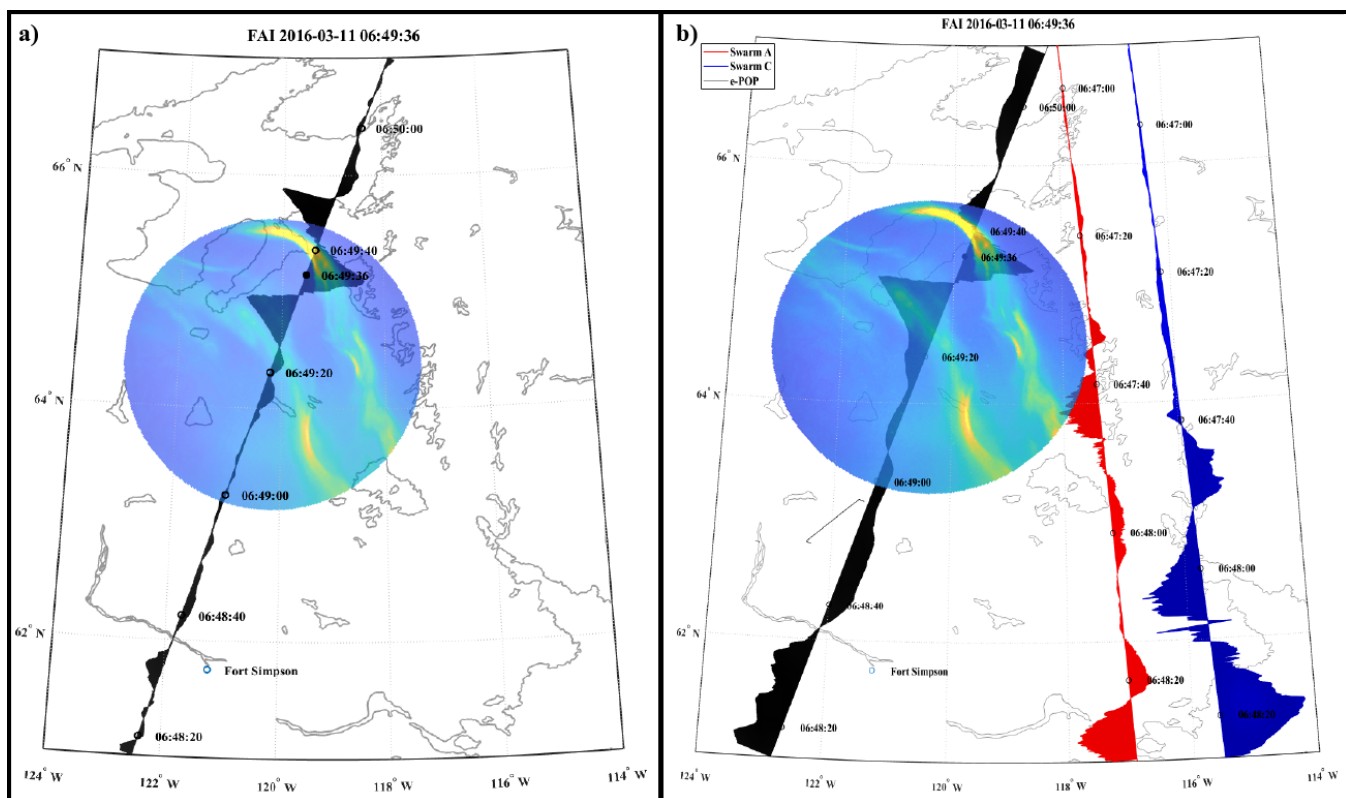

**Figure 6: a) Recreation of Figure 1 from Miles et al., 2018, but using the updated spacecraft attitude solution and vector calibration. Black shows magnetic variations in the outboard sensor NEC East component associated with the auroral currents visible in the superimposed image. b) Excerpted reproduction of Figure 1 from Miles et al, 2018 with font changed to improve visibility. Black shows the cross-track spacecraft $B_y$ component. The red and blue traces in b) show the cross-track magnetic field for Swarm A and Swarm C respectively reproduced from the original publication. Care should be taken when interpreting the image. The solid dot at 06:49:36 represents CASSIOPE's position when the overlayed image was taken. As such, only the portion of the magnetometer data that coincides with that dot is related to what was being detected.**

## 8 Future Work

The presented calibration is robust and a significant improvement to the static pre-flight calibration used previously. However, work on improving upon the results is ongoing. The work reported in this paper only focused on using the 12 standard calibration coefficients to improve the utility of the MGF data product. The housekeeping data from the BUS telemetry files will be used to identify when the various spacecraft subsystems turn on or off in an attempt to identify them as potential noise sources. However, the current dominant noise source is the reaction wheel tone (Finley et al., 2022). Once the wheels are mitigated for data prior to the first reaction wheel failure, we expect to be able to characterize and remove the stray fields resulting from various supply currents in the spacecraft and include those results in the next data release. The final goal for the next data release will be to regularize the calibrations using a lasso regularization (Tibshirani, 1996) which will

reduce the variability in the individual calibration results by penalizing outlier values based on an a priori model.. Looking further ahead, Ness-style gradiometer noise removal was intended as part of the initial mission design (Wallis, 2015); however, it was found to exacerbate the already significant reaction wheel tone. Once the wheel removal algorithms are operational (Finely et al., 2022), we hope to revisit using this as a potential method of noise removal We look forward to sharing additional improvements made to the data for the MGF instruments on Swarm-Echo.

## Code and Data Availability

The calibration software described in this manuscript is maintained in a GitLab repository at: https://research-git.uiowa.edu/space-physics/epop/mgftools. Daily MGF, Attitude, BUS Telemetry, and location files are publicly available, and can be obtained at https://epop-data.phys.ucalgary.ca/ .

## Author Contributions

RMB developed the Swarm-Echo calibration process and wrote the manuscript with contributions from all authors. The work presented here was developed at the University of Iowa under subcontract from the University of Calgary held and supervised by DMM. WH automated the improved CASSIOPE attitude solution and managed the production of Swarm-Echo data. ADH manages Swarm-Echo science operations and provided the Auroral imaging data.

## Conflicting Interests

The authors declare that there are no conflicting interests.

## Acknowledgements

The authors wish to thank Christian Siemes for developing the initial algorithm for the improved e-POP attitude solution, Chris Piker for automating the Swarm-Echo data product software, and Martin Rother for his analysis of the Swarm-Echo data and suggestions for improving the calibration.

## Financial Support

This research has been supported by the European Space Agency's Third Party Mission Programme.

## 9 Common Acronyms

ADCS: Attitude Determination and Control System

CASSIOPE: CAScade Smallsat and Ionospheric Polar Explorer

CDF: Common Data Format

CRF: Common Reference Frame

CSS: Coarse Sun Sensor

Dst: Disturbed storm time

e-POP: enhanced Polar Outflow Probe

ESA: European Space Agency

FAI: Fast Auroral Imager

ITRF: International Terrestrial Reference Frame

Kp: Planetary Index

MGF: Fluxgate Magnetometers

NEC: North East Center reference frame

SQUAD: Spherical QUADrangle Interpolation

YPR: Yaw Pitch Roll

µASC: micro Advanced Stellar Compass

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
