# Peer review of "In-Situ Calibration of the Swarm-Echo Magnetometers"

_EGUsphere, 2022_

## Author Comment (AC2)

Response to interactive comment RC1 on egusphere-2022-59 – "In-Situ Calibration of the Swarm-Echo Magnetometers" by Robert M. Broadfoot et al. by Mark Moldwin on 19 Apr 2022

*We would like to thank the referee for the constructive comments and suggested changes. Mark Moldwin suggested several excellent changes to the manuscript and raised important questions that we address below. Referee comments are in plain text, our response is in italics, and any changes made to the manuscript are in "quoted italics".*

This paper describes an in-flight calibration method for the CASSIOPE/e-POP, now known as Swarm-Echo, satellite that was launched in 2013 and included two fluxgate magnetometers on a shared boom. Several issues with the attitude determination system, failure of reaction wheels over time, and the natural drift of off-set and gains of the fluxgates contributed to the magnetometer data becoming less reliable. The paper describes applying a method to use the Earth's model geomagnetic field during quiet times and some rules on when to trust or discard attitude determination estimates to create a new "clean" magnetometer data set.

This is a useful paper describing the new calibration methodology and will enable expanded scientific use of the SWARM-ECHO magnetic data set.

Specific and General Questions interspersed below.

1. Line 11: "calibration performed between on data from January 3, 2014, to January 30, 2021"

   *This was a typo missed in our initial review, the text now reads:*

   *"Here we present the results of an in-situ calibration performed on data from January 3, 2014, to January 30, 2021."*

2. What is the length of the boom and the distance to the two magnetometers?

   *Boom length is 92.9 cm from center of hinge to end. There is 32 cm of separation between the two magnetometers center-to-center.*

   *Text was added on line 24 to address the length of the boom and magnetometer separation:*

   *"…separated by 32 cm center-to-center on a 92.9 cm carbon fiber boom (Figure 1)."*

3. Line 20, for comparison – what is the orbital altitude of e-POP compared to the other SWARM spacecraft? Are there any "conjunctions" that can be used to calibrate magnitude (and with field-line tracing) the direction of the field between the SWARM spacecraft?

   *The original Swarm spacecraft have roughly circular orbits at ~450 km for A and C and >500 km for B. e-POP was considered to be a desirable addition to the Swarm constellation because the orbit is highly*

*elliptical with perigee at ~330 km and apogee at ~1500 km at launch. As such, it sweeps out a broader range of altitudes that Swarm A/B/C. However, there are >3000 conjunctions between Swarm-E and Swarm-A/B/C where the spacing between Swarm-E and another Swarm satellite is less than 400 km, which we expect we will be able to make use of for further calibration and scientific purposes in the future.*

*Currently, the MGF residuals have an average rms error of ~10 nT compared to the Chaos model compared to the Swarm VFM average residuals of ~3 nT making MGF the dominant noise source. Until the MGF residuals are calibrated down below the Swarm VFM/Chaos residual level the before field-line tracing offers a benefit that overcomes the spatial coverage advantage of calibrated against a model field.*

*Text has been appended to the end of the paragraph starting at Line 20 to inform the reader of the difference in orbits, it reads:"The original Swarm spacecraft have roughly circular orbits at ~450 km for A and C and >500 km for B. e-POP was considered to be a desirable addition to the Swarm constellation because the orbit is highly elliptical with perigee at ~330 km and apogee at ~1500 km at launch. As such, it sweeps out a broader range of altitudes that Swarm A/B/C."*

4. Line 34: Suggest breaking last clause into separate sentence…" attitude. However that time interval is beyond the scope of this manuscript."

   *We have implemented this correction, and the line now reads:*

   *"…the spacecraft into a permanent spin-stabilized sun-pointing attitude. However, that time interval is beyond the scope of this manuscript."*

5. Line 65, 125: "data have…"

   *Line 65 now reads:*

   *"We assume that the raw sensor data have error in offset (**b**)…".*

   *Line 125 now reads:*

   *"Early mission attitude data were generated only as Yaw-Pitch-Roll (YPR) values using…".*

6. Line 135: What is SQUAD/SLERP?

   *Slerp stands for spherical linear interpolation and refers to interpolating between two orientations by moving at constant speed along a circular arc (as opposed to linear interpolation or lerp which idealizes the change in orientation by using linear polynomials and tends to result in very large*

*angles of arc between each interpolation point). This is sufficient if there is a small enough change between orientations and can treat each interpolation between two points as isolated, or either you don't know or don't really care if other changes in orientation exist before or after.*

*SQUAD or Spherical QUADrangle interpolation is a series of slerp interpolations that assumes that other orientations exist before, during, and after the current interpolation between the two points. It smooths the connections between the interpolations as it does not treat each interpolation as an isolated event. This has roots in computer animation and is useful for smoothing movement between multiple animation frames. This of course translates well to spacecraft attitude as the craft is constantly changing orientation while moving along the orbital path.*

*The line has been edited to improve clarity and correct one of the acronyms, and now reads:*

*"Switching to a Spherical linear interpolation (Slerp) or a Spherical QUADrangle (SQUAD) (Shoemake, 1987) interpolation rather than per-element interpolation further improved the robustness of the attitude transform by enforcing continuity and smoothness of the attitude solution over multiple measurement points which is appropriate for a physical spacecraft moving in physical space ."*

7. Line 151; Is "&" used intentionally instead of "and"?

   *That is simply a mistake made when transcribing the notes from one of the co-authors. The line now reads:*

   *"Following this, attitude solutions derived from the coarse sun sensors and bus magnetometer are considered."*

8. Line 164: "metadata are…"; Line 165: "and are included"

   *Line 164 now reads:*

   *"Supplementary metadata are also derived from these definitive solutions…".*

   *Line 165 now reads:*

   *"…the raw data source to be measured and are included…".*

9. A lot of the work is attempting to get a good handle on the attitude of the spacecraft despite the loss of sensitivity of the star trackers and other ADS efforts. Is there housekeeping information that tells you when different subsystems are on or off to attempt to assess the magnitude of the spacecraft noise? What is the relative magnitude of the residual pointing accuracy error on the final data product compared to your estimate of the spacecraft noise?

*Yes there now is a publicly available housekeeping data and plots that shows what instruments were active and at what particular times. There is also a public BUS telemetry file that contains flags for when other various spacecraft subsystems turn on or off as well as the current supplied to them. Both of these are aiding us in our current quest to assess the impact of other spacecraft and instrument subsystems as potential noise sources. However, the current dominant noise source is the reaction wheel tone (Finely et al., 2022). Once the wheels are mitigated we expect to be able to characterize and remove the stray fields resulting from various supply currents in the spacecraft.*

*We have added text to address this point on lines 329-334 it reads:*

*". The housekeeping data from the BUS telemetry files will be used to identify when the various spacecraft subsystems turn on or off in an attempt to identify them as potential noise sources. However, the current dominant noise source is the reaction wheel tone (Finley et al., 2022). Once the wheels are mitigated for data prior to the first reaction wheel failure, we expect to be able to characterize and remove the stray fields resulting from various supply currents in the spacecraft and include those results in the next data release"*

10. Line 192: "…important than the quantity" (than instead of that)

    *That line now reads:*

    *"We have found that the quality of data selected to derive the in-situ calibrations is generally more important than the quantity of data."*

11. Line 208: What is the effect of saturation of the sensor heads in terms of calibration? Was this a big effect initially (compared to the ground-calibration values), but once sensors were repeatedly permed up on orbit, minimal effect? (From Table 1 there seems to be essentially no impact on Gain.).

    *Our use of the word 'saturation' was unfortunately misleading. The toward analog path is clipped by the stray field from the magnetorquers (Wallis, 2010). The MAG's offsetting design has a limited instantaneous bandwidth which is temporarily clipped when the magnetorquers suddenly generate a 6000 nT step at 1m as the magnetic feedback cannot slew fast enough to compensate. We have updated the text to reflect this. It now reads:*

    *"From the Bus telemetry files we flag any data where the magnetotorquers were engaged as they suddenly produce a 6000 nT step of stray field on axis at 1 m (Wallis, 2010) which exceeds the bandwidth of MGF, rendering the data clipped and unusable."*

12. Table 1: Though having a large stray field from the boom makes sense since the large X offset is seen in the outboard sensor and not the inboard, what is the boom made from that could give such a

large field? It would be of interest to see the pre-flight off-set values to get a sense of the combination of the spacecraft fields and off-set drift combined.

*Compared to the preflight zeros, the offsets found in-situ are significantly larger and are not consistent with a decaying stray dipole field from the spacecraft. The in-situ offsets are also quite stable with time so they do not seem attributable to instrument aging. The boom is primarily carbon fiber with non-magnetic fasteners. However, there is a survival heater that was placed near the outboard end of the boom during integration, which is a potential culprit despite being un-powered. This is especially unfortunate because the offset is so large, it results in a proportionally larger rms error for the outboard sensor. When the reaction wheel tone is removed, the rms error for the outboard sensor is consistently larger than inboard.*

*We have added the pre-flight zeros to Table 1 along with text on lines 254-259 explaining how and why they are different from the offsets. It reads:*

*"The calculated pre-flight zeros (Wallis, 2010) are included for completeness. However, it should be noted that these zeros are not the same as the offsets calculated as part of the in-situ calibration as they only represent the intrinsic zero-offsets of the sensor and electronics and do not include effects from stray field from the as it was not possible to quantify those prior to integration. The large offset in $b_1$ for the outboard sensor is likely due to a magnetized object on the boom near the sensor – the survival heater added to the boom during final integration is a potential candidate."*

*Additionally, the description for Table 1 was updated and the following lines were added:*

*"The pre-flight values marked with '\*' represent the instrument zeros, or the offsets that would be measured by the magnetometer in zero magnetic field and are not the same as the in-situ offsets which include effects from stray-field from the spacecraft."*

13. Line 250: "taken"

   *That line now reads:*

*"This implies that the reaction wheel tone has a significant impact on the calibration results and that steps will need to be taken to mitigate the wheel tone in future data releases and doing so should have a significant impact on the calibration results for the early mission."*

14. Figure 5. Is this "all" the data or only Kp<3 and small change in Dst "quiet" data?

   *We double checked the code for the generated plot and noticed that it did not include the Kp and Dst flags as those are normally kept separate from the finished product. We have corrected this and updated Figure 5.*

15. Line 279: "….in in-situ…"

    *That line now reads:*

    *"…legitimate changes in in-situ geophysical field."* .

16. Was the inboard and outboard sensor used in a "Ness"-type gradiometer way to remove any spacecraft noise? If not, why not?

    *Not currently. Ness-style gradiometer removal was intended as part of the mission design; however, it was found to exacerbate the already significant reaction wheel tone – suggesting that the sensors are in the near-field of the wheel tone where a simple dipole approximation is insufficient.*

    *We have tested different algorithms, two of which (simple differencing and a notch filter) were not magnitude persevering or had difficulty adapting to changing wheel rates. There is a paper now in pre-print detailing the use of Multi-Channel Single Spectrum Analysis (M-SSA) at https://www.essoar.org/doi/abs/10.1002/essoar.10511290.1 . This has shown very promising results.*

    *The use of a Ness-style gradiometer will be revisited once the reaction wheel mitigate algorithms (Finley et al., preprint) are operational.*

    *We have addressed this point in Section 8, and added the corresponding citations to the references.:*

    *"Looking further ahead, Ness-style gradiometer noise removal was intended as part of the initial mission design (Wallis, 2015); however, it was found to exacerbate the already significant reaction wheel tone. Once the wheel removal algorithms are operational (Finely et al., 2022), we hope to revisit using this as a potential method of noise removal."*

17. Figure 6. What are the red and blue traces in panel b?

    *The red and blue traces represent the cross-track measurements from Swarm-A and Swarm-C which passed through the same region of interest at a different time, which is useful from a scientific perspective. Those were not included in the recreation as the main goal was to show the change in the e-POP measurements. As a follow-up, it will be interesting to revisit the analysis done in the original paper and see what if anything has changed with the recent calibration and attitude updates.*

    *We added a note about the Traces in the description for Figure 6 it reads:*

    *"The red and blue traces in b) show the cross-track magnetic field for Swarm A and Swarm C respectively reproduced from the original publication."*

18. A wild idea not necessarily to pursue for this study (following on the "conjunction" idea in a statistical sense given above), is to use the other SWARM satellites to determine the magnetic equator (when the field is horizontal) and compare locations of Echo with the other SWARM for different 7 day intervals. The poles do shift and move over months/years, but the equator should be pretty "fixed" over multiple 7 day intervals allowing for Echo to pass over the same longitude sector. The offset in location potentially can be used to estimate "off-set" in angle using the CHAOS field line mapping.

*This does seem like an interesting idea to try in future work. In the immediate future, we're working to remove time-varying local field (observable from the difference between the inboard and outboard sensors) and minimizing our residuals to the limit of the Chaos magnetic field model as discussed in point 3.*

---

## Author Comment (AC3)

Response to interactive comment RC2 on egusphere-2022-59 – "In-Situ Calibration of the Swarm-Echo Magnetometers" by Robert M. Broadfoot et al. by Kenneth R. Bromund on 13 May 2022

*We would like to thank the referee for the constructive comments and suggested changes. Kenneth R. Bromund raised several important questions and suggested several excellent changes that are addressed below. Referee comments are in plain text, our response is in italics, and any changes made to the manuscript are in "quoted italics".*

General Comments
* * *
This is an important paper describing the methods used to calibrate the processed CASSIOPE fluxgate magnetometer (MGF) data products that are currently available to the public.

The results are impressive.   However, the style is uneven and at times there are errors, omissions or inconsistencies that must be remedied in order to make the descriptions of the  methods sufficiently accurate for publication.

Specific Comments
* * *
Section 1:

Line 49:  Tests were run to evaluate mutual interference of the two sensors.  Was any interference identified?  If so, how is this information used to aid the calibration process?

*Information from the Wallis 2010 technical report on the MGF pre-flight calibrations was added to Section 2 and the text re-arranged slightly to improve the flow. It provides additional information regarding the tests for mutual interference and the results of those tests.  the text starts on line 59 and ends on line 64, and reads:*

*"Additional pre-flight tests were completed to estimate temperature dependence and the mutual interference of the two sensors. Mutual interference was found to exist between the two sensors due to the modest 32 cm of separation between them. On CASSIOPE, the instruments share a common clock which mitigates any interference from the respective drive signals making the interference primarily one of gain error from the stray fields generated by each sensor's magnetic feedback. The prefight interference was characterized by calibrating the sensors against a variety of fields while powering the other sensor on and off independently. Maximum interference was found to be in the +Z direction (Wallis, 2010) with the presence of a second sensor causing an error of approximately $-3.6x10^{-4}$*

*nT/nT. On-orbit however, the sensors are never operated independently, so this effect is always present and is intrinsically captured by the instrument gains and then corrected by the application of the in-situ calibration.*"

Section 3:

Although Olsen et al. (2003) is referenced for the notation used in this paper, there are many departures from this notation in this section, and it is unclear how the notation in this section relates to the results presented in Section 7.

*Excellent point - this was our oversight when writing the results section. The notation in section 7 has been updated to be consistent with the notation in section three and be generally consistent with Olsen et al., 2003.*

*As for the departures from notation, the calibration is based on Olsen et. al, 2003 but does not follow it exactly as Ørsted contains an absolute reference for calibration whereas Swarm-Echo does not which does alter the calibration process slightly.*

Line 67:  it would be more accurate and consistent with Olsen et al (2003) to say that the raw sensor data, E, is in engineering units that are approximately equivalent to nT.
*Agreed. That line now reads:*

*"Let **E** be the raw sensor data (in engineering units, eu) that is related to the magnetic field vector…".*

line 71,  As used in Equation (1),  b is a pseudo-vector (in a non-orthogonal system), and in engineering units.  specifying 1,2,3 is appropriate for the non-orthogonal system, but I note that this is inconsistent with Table 1...  do the results offX, offY, offZ in Table 1 correspond to b1, b2, b3 (in which case, they might be labeled as engineering units, not nT), or are they actually nT (ie they are the offsets to be subtracted in the orthogonal, calibrated system)?

*This is addressed in later comments, but off(X,Y,Z) have been updated to match the notation of section 2. This is technically correct as $\tilde{\boldsymbol{b}}$ is technically in units of nT as it is applied after the calibration matrix, however **b** is not. As no preflight calibrations have been applied to the measurements the values are in eu. The table has been updated to reflect this.*

*The text describing the parameters has been updated and now reads:*

*Line 67:*

*"Let **E** be the raw sensor data (in engineering units, eu) that is related…"*

*Line 71:*

*"...is a vector of offsets (given in units of eu) comprising the superposition..."*

Line 74: It would be appropriate to describe the dimensions of S as eu/nT, consistent with the results in your Table 1 in section 7. Subscripts 1,2,3 would be more appropriate, as S is in the non-orthogonal system...
*The text and equations have been updated to reflect these comments*

Lines 78-79, If Equation (4) is correct, then the angles u1, u2 and u3 are not the same as the angles of the same names in Olsen et al. (2003). Please describe how u1, u2 and u3 defined, and their relation to the angles Oxy, Oxz, and Oyz reported in Table 1. Are they the angles between each pair of the slightly non-orthogonal sensors 1, 2, 3? While a figure would be helpful, it would be sufficient to define the terms clearly in the text, indicating, for example, that P represents the projections of sensors 1, 2, and 3 onto the orthogonal magnetometer reference frame, that sensors 1 and 2 are presumed to be in the X-Y plane, etc... Consider using a different notation (e.g. o12, o13, o23 ) that clarifies that these angles are not same as u1,u2,u3 in Olsen et al. (2003).

*Excellent point - we checked the mathematics associated with our version of the orthogonality matrix and found a sign error in the $P$ and $P^{-1}$ matrices second row, first term (See included correction document). This will change the sign of the obtained orthogonality values and trended parameters, but not the overall conclusions drawn since the actual calibration is performed using the fit coupling matrix rather than its decomposed physical parameters. To better reflect the Olsen (2003) document, the matrices for $P$ and $P^{-1}$ have been changed to match that document and now show as:*

$$\boldsymbol{P} = \begin{vmatrix} 1 & 0 & 0 \\ -\sin(u_1) & \cos(u_1) & 0 \\ \sin(u_2) & \sin(u_3) & \sqrt{1-\sin^2(u_2)-\sin^2(u_3)} \end{vmatrix} \quad (4)$$

*And*

$$\boldsymbol{P^{-1}} = \begin{vmatrix} 1 & 0 & 0 \\ \dfrac{\sin(u_1)}{\cos(u_1)} & \dfrac{1}{\cos(u_1)} & 0 \\ -\dfrac{\sin(u_1)\sin(u_3)+\cos(u_1)\sin(u_2)}{w\cos(u_1)} & -\dfrac{\sin(u_3)}{w\cos(u_1)} & \dfrac{1}{w} \end{vmatrix} \quad (7)$$

*With w defined as:*

$$w = \sqrt{1-\sin^2(u_2)-\sin^2(u_3)}$$

Lines 79-80. The definitions of the Euler angles (order 1-2-3) appears to differ from Olsen (2003), can you provide a reference? A more detailed description would be helpful to evaluate the physical significance of each parameter, when evaluating the results presented in Section 7.

*The choice of rotation is a convention chosen by individual spacecraft teams and in some ways is arbitrary as long as it fully describes the rotation. There are other ways that the rotation can be described, however we chose to keep the convention determined by the original CASSIOPE team.*

*Text was added to the document to make note of this discrepancy on lines 79-80 it reads:*

*"It should be noted that the choice of rotation order is largely arbitrary as long as it is able to fully describe the rotation from the instrument reference frame into the CRF."*

Section 4:

There is discussion of the "original" attitude system based on STK, but it is not clear what is the relevance of this system to this study.  Was it relevant to previously released versions of the data?  If so, please specify.

*The purpose of this paper is to detail the entire process of improving the quality of the MGF data and one of those is an update to the attitude determination software. Lines 132-143 detail the issues with using our old method. An introduction was added to section 4 to explain the need to for an accurate attitude solution for vector calibration as previously it jumped into describing the methods of attitude determination without context. The attitude solution based on our old method was used for versions 1.x.x of the data product whereas the improved solutions were used for 2.x.x. The current attitude file version of 1.3 should not be confused with the data product version.*

*The introduction is on lines 131-136 and reads:*

*"An accurate attitude solution is critical to a successful vector calibration. It is used to rotate the reference magnetic field from a geophysical coordinate system into the coordinate system of the magnetometers. Though CASSIOPE has several methods to determine spacecraft attitude, the difference between the quality of attitude sources is large. Deficiencies were found in the original attitude determination software that impacted the overall quality of the attitude solution. This section will focus on the updates that were made to the attitude determination software with Section 5 detailing the method used to select quality attitude data and rotate the reference magnetic field into the magnetometer's coordinate system. The previous attitude solution was used for versions 1.x.x of the data product whereas the improved solutions were used for 2.x.x. The current attitude file version of 1.3 should not be confused with the data product version."*

What were the methods used to verify and obtain the revised attitude solutions that "included improved alignment between different star camera modes, corrections for chromatic aberration and thermal effects in the star cameras, and corrections to frame, location, and epoch transforms"?

*The details of this will be detailed in an upcoming paper by two of the co-authors, Warren Holley and Andrew Howarth, with a publication time targeted for October of this year. However, while the details are beyond the scope of this paper, here are some sources to answer those specific questions:*

*The thermal correction method is based on a technical report by Dr. Christian Siemes which is ESA-EOPSM-SWRM-TN-3487. From this, they used his methods to update the thermal correction function and relied on different functions for the rest of the processing code mentioned in this document. We have made this technical report available at* [https://epop.phys.ucalgary.ca/wp-content/uploads/2022/06/ESA-EOPSM-SWRM-TN-3487_New_Star_Sensor_Attitude_Solution.pdf](https://epop.phys.ucalgary.ca/wp-content/uploads/2022/06/ESA-EOPSM-SWRM-TN-3487_New_Star_Sensor_Attitude_Solution.pdf)*.*

*The use of chromatic aberration in our manuscript was incorrect as the corrections are instead for orbital and annual correction. These are caused by the relative perpendicular velocity of the stars with the camera heads. Orbital aberrations are dependent on the spacecraft velocity in the ICRF frame while annual aberrations depend on the Earth's velocity around the sun. See page 83 of the Swarm Level 1b Processor Algorithms* [https://earth.esa.int/eogateway/documents/20142/37627/swarm-level-1b-processor-algorithms.pdf/e0606842-41ca-fa48-0a40-05a0d4824501?version=1.0](https://earth.esa.int/eogateway/documents/20142/37627/swarm-level-1b-processor-algorithms.pdf/e0606842-41ca-fa48-0a40-05a0d4824501?version=1.0) *.*

*It should be noted that section 6.4.2.3 is the equivalent of the thermal correction method developed by Dr. Siemes, but his only uses two cameras.*

*Epoch correction is correcting for the onboard clock drifting in its time measurement. To do this we keep track of the GPS "Pulse Per Second" time-sync transmission to correct both clock speed and absolute time. This can only be done once per day. These corrections ARE rare but occasionally a coarse adjustment does need to be made (usually by integer seconds) to the 'raw' timestamps stamped to the telemetry packets.*

*The historic frame transforms used were not sufficiently accurate for this purpose. The issue was that originally the merged attitude solution (in j2k/ICRF, more on this in a later comment) determined by the spacecraft was used to calculate the yaw pitch and roll angles without any further processing and correcting. The issue with the onboard merged solution is that it cannot be decoupled back into individual star-tracker solutions, so the corrections mentioned above could not be applied. Without these corrections, and without distinguishing what solutions were being fed into the processor, the quality and accuracy of the attitude solution began to degrade as the mission progressed.*

*The word chromatic has been removed from the text and replaced with orbital and annual along with a small description of this method and the corresponding reference was added to text. Additionally, a note to the readers about the forthcoming paper was also added.*

*The text begins on Line 134 and reads:*

*"…corrections for orbital and annual aberration which are caused by the relative velocity of the stars within the camera heads and thermal effects in the star cameras. These corrections closely follow the procedures outlined in the Swarm Level 1b Processor Algorithms (SW-RS-DSC-SY-0002). Additionally, corrections were made to frame, location, and epoch transforms. The details of these corrections will be covered in a forthcoming publication by the CASSIOPE science operations team. "*

Is the uASC accuracy of <2 arcseconds achieved only the beginning of the mission, or is this the lifetime accuracy specification?  How was this verified?

*This is based on the end-of-life estimates given by the manufacturers which is detailed in the reference provided in the manuscript (Jorgenson et al., 2003). An additional source for estimated accuracy is* [https://articles.adsabs.harvard.edu//full/2004ESASP.571E..55J/0000055.002.html](https://articles.adsabs.harvard.edu//full/2004ESASP.571E..55J/0000055.002.html)*. The accuracy as well as projected lifetime can be found in Table 1, column 3.*

*Additionally, the mentioned reference has been changed as it did not previously include et al.,*

*We have updated the text to include the point about the end-of-life estimate and included et al. in the reference. It now reads:*

*"The µASC is specified by the manufacturer to provide solutions with errors less than 2 arcseconds (3σ) until end of life (Jørgensen et al., 2003), though there is a known mounting angular offset of ~0.447 degrees to bring the two cameras into agreement, which is averaged across both cameras."*

Line 135:   At this point, it is enough to know that "improved methods of attitude interpolation are applied to achieve robust attitude transformations",  and leave the detail about SQUAD/SLERP to section 5, where you provide the reference...

*This has been changed in the text, the reference is now introduced on line 142 and the text has been altered on lines 141-142 to improve the clarity and now reads:*

*"Switching to a Spherical linear interpolation (Slerp) or a Spherical QUADrangle (SQUAD) (Shoemake, 1987) interpolation rather than per-element interpolation further improved the robustness of the attitude transform by enforcing continuity and smoothness of the attitude solution over multiple measurement points which is appropriate for a physical spacecraft moving in physical space."*

The relationship between section 4 and section 5 is confusing.  In some ways, section 4 appears to be intended as an introduction to section 5... It would be helpful to mention which topics will discussed further in section 5, and which topics are beyond the scope of this paper.

*The intention of section 4 is to introduce the fact that the attitude solution needed to be updated prior to beginning the in-situ magnetometer calibration and the reasons for it. Section 5 is intended to detail the process used to select quality attitude data and the process needed to rotate the reference field into the magnetometer coordinate system. The new introduction described above is intended to help clarify this distinction in Section 4. We have made several other clarifications to Section 5 as described below.*

Section 5:

I feel like I should be able to understand exactly what this section is talking about, but I'm finding it extremely difficult to follow.

*In re-reading the start of Section 5 we noticed that the title is misleading and have revised it to:*

*"Attitude selection and reference field rotation"*

*This, combined with the introduction in section 4 should help clarify the intent and content of these two sections.*

Lines 138-139:  This sentence is unclear.  I think it's a typo. Is it intended to say:

"In-situ calibration of the MGF instruments requires rotating the reference magnetic field from its native North, East, Center (NEC) frame into the local CRF frame of the spacecraft by convolution against the spacecraft attitude solution." ?

*This is correct, and the text now reads:*

*"In-situ calibration of the MGF instruments requires rotating the reference magnetic field from its native North, East, Center (NEC) frame into the local CRF frame of the spacecraft by convolution against the spacecraft attitude solution".*

Line 143-145:  The wording of this sentence is very confusing, and I'm pretty sure I don't understand it correctly....  "The coordinate system that defines this spacecraft (SC) coordinate system..."  I'm used to thinking of SC coordinates to mean a system fixed to the spacecraft, and the reference to Figure 2, in particular, appears to re-enforce that assumption:  It is the same figure that is used to illustrate the CASSIOPE spacecraft frame (CRF?) in the CASSIOPE DATA HANDBOOK online...   Meanwhile, this sentence appears to describe a coordinate system defined by the spacecraft orbit:  ie. one in which "+X points  towards ram [and perpendicular to nadir], +Z points nadir, +Y completes the right-handed system".  This is what the CASSIOPE DATA HANDBOOK calls the Orbital Reference Frame... is this what you are calling the SC coordinate system, in this paper?    If so, then it does make sense to say that SC and CRF are co-aligned when YPR are all zero, as implied by this sentence....

*We did not reference the handbook in our manuscript as it is still a work in progress and will be undergoing changes prior to the official release to deal with these inconsistencies. The upcoming paper mentioned above in the comments for Section 4 will provide more insight into this question. To clarify however, CRF, body, and SC are all supposed to be the "same" frame of reference and are used as such in the Swarm documentation. However, due to uncertainties in the frames, SC is the physical frame of the spacecraft, CRF is a synthetic frame that is equivalent in definition to SC but includes known offsets due to uncertainties in the merged attitude solution. Orbital or ORF is idealized Nadir with YPR = 0 as defined by the J2k/ICRF ephemeris data. SC and CRF are aligned coarsely, but not perfectly, at all times. Since the star-trackers are hard-mounted on the body, any drift in YPR will be represented, and theoretically equal, in both CRF and SC (Body).*

*That being said: the attitude solution we work with is CRF as it has uncertainty, and any frame transforms done will include that uncertainty so for this manuscript is a distinction without a meaningful difference. We have removed all mention of SC or body and replaced it with CRF to improve the clarity of this section. CRF is now defined initially in Figure 1.*

*The text for Figure 1 now reads:*

*"Figure 1: The CASSIOPE spacecraft showing the two MGF magnetometers mounted at different distances from the spacecraft body on a common boom. The coordinate system shown represents the Common Reference Frame (CRF) with +Z pointing nadir, +X pointing ram, and +Y completing the right-handed coordinate system."*

Line 149:  What is the relevance of this 'secondary-source attitude solution' if you are are using the raw star camera data to get superior revised attitude solutions? Do you have access to the raw quaternions representing the attitude of the star camera frame? Wouldn't the raw output of the star cameras be quaternions representing the star camera frame in the ECI J2000 coordinate system?  How does J2000 get transformed to SC?  The transformation would require the ephemeris as an input.   Is this done on board, or is it re-calculated on the ground?

*The relevance of the secondary source attitude solution is that it exists and serves as a backup when the solutions are determined to be more accurate than one provided by the Star-Trackers (usually during camera dropouts). These solutions, though flagged and not used during calibration, still appear in the MGF data product with a flag warning the user of the potentially lower quality of the solution. The choice is then up to the user if the data that are associated with that solution are usable.*

*The mathematics of the transformations will be outlined in the paper mentioned in the Section 4 comments.*

*The satellite only uses j2k/ICRF for it's internal attitude control. It doesn't take location into account instead it takes the current time and the position of the stars. The j2k/ICRF solutions have a mounting quaternion that rotates them into CRF which is the raw solution that is transmitted. All additional merging and processing of the quaternions are done on the ground by science operations center. An onboard merged solution IS created, but is of poor quality compared to an individual solution or merged solution processed by science operations center since it cannot be decoupled and corrected using the corrections mentioned in the comment for section 4.*

*Information on the angular offset of the star trackers is in Section 4, Line 121 and issues with the onboard merged solution as well as when the secondary source attitude solutions are used are in Section 5 Lines 149-155.*

*A note informing the reader that the mathematics for the CRF-to-ITRF transformation will be detailed in the paper mentioned in Section was added to line 163.*

*A note informing the users of the inclusion of all data and flags in the final CDF data product was added starting Line 208.*

*Line 163 reads:*

*". The mathematics for the CRF-to-ITRF transformation will be detailed in the forthcoming publication mentioned in Section 4"*

*Line 208 reads:*

*"It should be noted that even though the flagged data are excluded from the calibration process, all data and attitude flags are included in the daily MGF data product so the user can decide if the flagged data are useable."*

Line 156: "the attitude solutions are then rotated into CRF":  rotated nto  CRF from what system?  Does this mean, for example, that you apply an X-to-CRF rotation appropriate to each attitude knowledge source (X could be "star tracker A", "star tracker B", etc.) to obtain a CRF to SC attitude solution?

*As mentioned in the response for lines 143-145, all mentions of SC or Body have been changed to CRF.*

*This line has been edited for clarity and now reads:*

*"The attitude solutions are then rotated from the coordinate frame of the source into CRF to bias interpolation towards …"*

Line 158:  SLERP is definitely to be recommended over per-element interpolation!   I would be concerned about splining with SQUAD unless you are sure that the precision of each solution is much smaller than the change in attitude from one solution to the next.

*Because CASSIOPE is a three-axis stabilized spacecraft, the precision of the attitude solution is smaller than the change between individual solutions at a 1-s cadence.*

Lines 158-161.  Confusing sentence.  Maybe SC serves as a refers to a specific coordinate system in the first instance, and as an abbreviation for 'spacecraft' in the second?

*We have replaced all mentions of SC with CRF.*

In flight, one can only calibrate against the attitude determination system's idea of what CRF is, which will always contain a bias with respect to the mechanical CRF that was used to measure alignments on the ground.   As you have noted, this bias will vary depending on the attitude source (star camera A, star camera B, etc.) and the solution will have varying levels of noise, depending on the source...   Have you used the vector-vector calibration to evaluate the relative bias of each source, and then incorporated these bias corrections into the Swarm-Echo attitude CDF product?  (That is what I assumed you meant back in section 4, when you mentioned 'improved alignment between different star camera modes'...)

*We have not attempted this yet. The improvements in the attitude solution, documented in the upcoming publication from the Cassiope Operations Team, aligned the attitude solutions generated by the different potential attitude solutions by fitting rotational offsets until the virtual transient caused by changes in attitude solution was minimized. The transform of this, now common, CRF to the geophysical*

*frame is then captured by the euler angle terms in the in-situ calibration. Using the vector-vector calibration to check the rotational offsets between different attitude solution inputs is an excellent idea for future work.*

Line 160: What is a Body-to-ITRF transformation?  How is it derived?  If Body = SC, then I would agree that a SC-to-ITRF is what is required at this point.  It would be well-defined, given the ephemeris data expressed in ITRF as an input.   A reference, or more details would be helpful.   The documentation of the Attitude Quaternion File in the CASSIOPE PROCESSED DATA HANDBOOK seems to use the term "ITRF<-Body" to refer to the final attitude quaterion itself, rather than something used rotate the interpolated quaternions at this stage.

*This was addressed in the comment for lines 143-145 above. All mentions of SC (body) have been changed to CRF.*

In any case, have you verified the accuracy of the ephemeris?

*The accuracy of the ephemeris was verified in Montenbruck et al., 2019 as well as mentioned in an announcement from the European Space Agency (https://earth.esa.int/eogateway/news/swarm-e-cassiope-precise-orbits-and-attitude-data-released ) and found to be within 10 cm when GAP-A was operating continuously and 1 m when interpolation was needed. During long down periods, extrapolation was used resulting in a larger error. It should be noted that the larger error in the attitude is referencing an older iteration of the solution not used in production, not the one used in this document.*

*Text has been added to line 179 informing the reader that the ephemeris has been validated and citing Montenbruck et al., 2019 it reads:*

*"Preliminary analysis suggests that the accuracy of the ephemeris is comparable to the values determined by Montenbruck et al., 2019."*

Lines 176-179:  Reading this, it seems that SC and CRF are actually supposed to be the the same thing...  So, either I misunderstand this paragraph, or I misunderstood everything leading up to it...

*This was addressed in the comment for lines 143-145 above. All mentions of SC (body) have been changed to CRF as they represent the same coordinate convention.*

Line 180: What is SP3 format?

*SP3 or .sp3 format is a precise ephemeris format associated with GPS. It was developed by the National Geodetic Survey (NGS) to store orbit information.*

*We have added text to the document on starting on line 180 to that effect:*

*"The daily attitude files are in CDF format and contain the $q_{ITRF \to CRF}$ quaternions and the daily ITRF position files are in SP3 format which is a precise ephemeris format developed by the National Geodetic Survey to store orbit information. Both are publicly available at [https://epop-data.phys.ucalgary.ca/](https://epop-data.phys.ucalgary.ca/) ."*

Section 6:

The data selection methods described are all reasonable.  The weighting method described in lines 220-228 seems appropriate, and is well referenced...

*Thank-you*

Section 7:

Table 1 and Figure 4: see my comments in section 3, regarding consistency of notation...

*Table 1 and Figure 4 have been updated so the notation now matches section 2.*

Line 256: what is meant by 'regularization'?

*Regularization refers to Lasso regularization. It is used to improve the accuracy of the fit and reduce variability between the individual calibration results by penalizing outlier values based on an a priori model. This will change equation 11 to match the form of equation 17 in Olsen et. al, 2020. For more specific information on the process, see Tibshirani, 1996.*

*We have added a note in section 8 lines 301-302 briefly describing this process as well as have included the corresponding reference.*

*". The final goal for the next data release will be to regularize the calibrations using a lasso regularization (Tibshirani, 1996) which will reduce the variability in the individual calibration results by penalizing outlier values based on an a priori model."*

Fig 4: Do you have an explanation for the greater variability observed  on the outboard sensor, as opposed to the inboard sensor?

*The larger variability in the outboard sensor is likely due to the large offset seen in the x, and to a lesser degree z directions. As the numbers are much larger in comparison to the other results, the variation in those values is also proportionally larger.*

*Text was added starting on Line 248. It reads:*

*"The larger variability in the outboard sensor is likely due to the large offset seen in the $b_1$, and to a lesser degree $b_3$ components. As the numbers are much larger in comparison to the other results, the variation in those values is also proportionally larger. "*

What mechanism is assumed to be the cause of the reaction wheel tone in the MGF data?  Is it electromagnetic, or mechanical?  If mechanical, it might explain why the deviations in the parameters is similar for both the inboard and outboard and outboards sensors, and that the outboard deviations appear to be slightly larger in some cases (the boom may amplify the vibrations at larger distances... )  if it is assumed to be electromagnetic, I would expect the outboard deviations to be smaller...

*This is addressed in the comment for section 8. A paper is in prelease that deals with this topic specifically. The amplitude of the of the reaction wheel tone is smaller in the outboard sensor which is consistent with the coupling being magnetic.*

I note that for the outboard sensor, there is a significant correlation between variations in off_x and the e3 parameter...  Perhaps only one of them is physically changing...  What degree of angular error would correspond with an offset error of 15 nT?

*It is plausible that only one of the parameters (for example, the offset) is significantly changing. The apparent correlation is likely due to a combination of the need for some regularization (as discussed above) and that the variation due to the temperature dependence of the offsets or some time-varying local source such as solar panel/battery current is getting combined into multiple terms as it has not been corrected for in this iteration of the data. This would potentially lead to a non-physical variation in the matrix decomposition.*

*Using a rough calculation: A 15 nT offset in a 55000 nT field results in ~ 0.016 degrees or ~57 arcsec of angular error.*

Lines 249-250:  sounds plausible... Can you show an example of the reaction wheel noise?   Is it a monochromatic high frequency (whether electromagnetic or mechanical) that is aliased down into the MGF frequency range,  or is it more broadband?    What is the amplitude?

*This is addressed more in the comment for section 8 as a paper is now in pre-print that goes a bit more in-depth on the noise and method of removal.*

*Wheel removal before and after the first wheel failure in August 2016 pose different problems. Prior to 2016 the fundamental wheel rates were usually high enough (~15 Hz) to be spectrally separate from*

*geophysical phenomena such as Alfven waves. However, after the 2016 wheel failure, the reduced wheel rates (~1 Hz) overlap directly in frequency space requiring special care to remove.*

Section 8:

I very much look forward to seeing these further developments!

For mitigating the reaction wheel tone: I would need to see examples, but I could imagine that more specific methods could be applied to identify and remove the 1 sps samples that are impacted by the reaction wheels... but indeed this might still require longer bins to make up for the lost data.

*For the reaction wheel removal, we are in the process of implementing a Multichannel Single Spectrum Analysis (M-SSA) algorithm that has shown a lot of potential in the measurements prior to the first wheel failure. The paper on the topic is now in pre-print and can be read at:*
*https://www.essoar.org/doi/abs/10.1002/essoar.10511290.1*

*One thing to note is that successful use of the wheel removal algorithm requires the full cadence data. We use a robust linear fit to reduce the sampling rate from full cadence to 1 sps. However, even this tends to couple a reduced residual of the wheel tone via frequency beating which leads to other problems.*

*As for the longer bins, we are looking into potentially using one-month bins similar to Cryosat-2 (Olsen et al., 2020).*

Technical Corrections
* * *
Line 59-60:  typo, should be 'et al.'

*The line now reads: "Olsen et al., (2003)."*

Line 47:  typo, should be 'stimuli'

*The line now reads: "…and apply various stimuli…"*

Line 193: typo, should be 'more important than'

*The line now reads: "…calibrations is generally more important than…"*

Line 250: typo, should be 'steps will need to be taken'

*The line now reads: "…results and that steps will need to be taken to mitigate the wheel*…"

---

## Author Comment (AC4)

This document serves as a notification of a correction to the mathematics for equations 4 and 7. It was discovered that the matrix used to find the preflight calibration orthogonality and extract the orthogonality values for trending the in-situ calibration parameters contained an incorrect sign for the term in row 2, column 1. To minimize confusion, the matrices have been changed to match the form used in the Swarm l1b Processor Algorithms (SW-RS-DSC-SY-0002 Issue 6.11), Olsen et al, 2003, and Olsen et al, 2020.

It should be noted that this only affects the sign of the extracted parameters and not the calibration itself. The calibration uses the coupled matrix outlined in equation 9. It does not alter the conclusions drawn from the results nor does it change the scope of the next steps to be taken. In addition to the updated matrices, the values in Table 1 and Figure 4 have been updated to reflect this change.

Equation 4 has changed from

$$\boldsymbol{P} = \begin{vmatrix} 1 & 0 & 0 \\ \sin(u_1) & \cos(u_1) & 0 \\ \dfrac{\sin(u_2)\cos(u_3)}{\cos(u_1)} & \cos(u_2)\sin(u_3) & \cos(u_2)\cos(u_3) \end{vmatrix} \tag{4}$$

To

$$\boldsymbol{P} = \begin{vmatrix} 1 & 0 & 0 \\ -\sin(u_1) & \cos(u_1) & 0 \\ \sin(u_2) & \sin(u_3) & \sqrt{1 - \sin^2(u_2) - \sin^2(u_3)} \end{vmatrix} \tag{4}$$

Equation 7 has changed from

$$\boldsymbol{P}^{-1} = \begin{vmatrix} 1 & 0 & 0 \\ -\tan(u_1) & \dfrac{1}{\cos(u_1)} & 0 \\ \tan(u_1)\tan(u_3) - \dfrac{\tan(u_2)}{\cos(u_1)} & -\dfrac{\tan(u_3)}{\cos(u_1)} & \dfrac{1}{\cos(u_2)\cos(u_3)} \end{vmatrix} \tag{7}$$

To

$$\boldsymbol{P}^{-1} = \begin{vmatrix} 1 & 0 & 0 \\ \dfrac{\sin(u_1)}{\cos(u_1)} & \dfrac{1}{\cos(u_1)} & 0 \\ -\dfrac{\sin(u_1)\sin(u_3) + \cos(u_1)\sin(u_2)}{w\cos(u_1)} & -\dfrac{\sin(u_3)}{w\cos(u_1)} & \dfrac{1}{w} \end{vmatrix} \tag{7}$$

With line 90 now including the definition for w:

Where $w = \sqrt{1 - \sin^2(u_2) - \sin^2(u_3)}$ and $\boldsymbol{R}_A^{-1} = \boldsymbol{R}_A^{T}$ from the properties of rotation matrices.

Additionally, text has been added to the description of